# Sketch First, Scale Fast:
# On Efficient Inference-Time Scaling for Visual Generation

## Abstract

Diffusion models have exhibited exceptional capabilities in image generation tasks. However, due to their inherent stochasticity, the quality of the generated images varies across different settings and may not always be high-fidelity and accurately align with user requirements. To address this challenge, recent works begin to focus on enhancing human preference alignment and improving overall image fidelity during the inference process with inference time scaling, which involves generating multiple candidates through repeated sampling and selecting with predefined metrics. Although effective, it will introduce considerable computational extra costs with the redundant sampling steps. To overcome these limitations, we propose a novel inference time scaling framework SF2 that enables estimators to make decisions in the early step via Co-Fusion pipeline, significantly improving the whole process while maintaining the quality of the selected images. In addition, for the continue generation part, we propose vision reflection mechanism to further align and correct the images with user requirements. Numerous experiments demonstrate that our proposed method can achieve comparable performance while $2.2 \times$ and $2.0 \times$ accelerating the whole inference time scaling process on both Stable-Diffusion-3.5-Large and FLUX.1-dev models.

## 1 Introduction

In recent years, diffusion models have demonstrated impressive capabilities across various generative tasks and have been widely adopted in diverse modalities, *e.g.*, image Ho et al. (2020); Song et al. (2020); Rombach et al. (2022); Esser et al. (2024); Labs (2024); Podell et al. (2023), video Ho et al. (2022); Blattmann et al. (2023); Kong et al. (2024); Jin et al. (2024); Chen et al. (2024b); Zheng et al. (2024), 3D Luo & Hu (2021); Poole et al. (2022); Deng et al. (2023), and text Gong et al. (2022); Chen et al. (2023). However, diffusion models inherently rely on stochastic processes, specifically involving randomly sampled initial noise, resulting in variability and inconsistency in output fidelity. Recent studies Zhou et al. (2024); Ahn et al. (2024); Qi et al. (2024); Bai et al. (2024) further reveal that the effectiveness of these initial noise vectors various markedly, often causing generated images to deviate from user requirements and do not consistently achieve high-quality outcomes.

In this case, a straightforward way to address this issue is to scale the training process Mei et al. (2024); Li et al. (2024); Liang et al. (2024); Podell et al. (2023); Chen et al. (2025) from aspects of training data volume, model size, and training duration, to enhance the generative capacity of models, and thereby reducing the probability of generating the undesirable output. However, training time scaling methods typically incur prohibitive training costs, and usually come with the issues including costly data acquisition and heavy computational resources demands. Consequently, recent research has turned to the inference time scaling method Ma et al. (2025); Xie et al. (2025); Kim et al. (2025), with the aim of obtaining at least a high-fidelity image with only a trivial increase in inference overhead compared to training costs. To retrieve outputs that align with user specifications, a simple, yet effective inference time scaling method performs a repeated sampling process to form a set of candidate images, from which the highest quality result is selected according to predefined evaluation metrics Ma et al. (2025); Xie et al. (2025). However, this method is inefficient and time-consuming, as it requires repeated complete passes through the entire image generation pipeline to obtain a sufficient large candidate set. Given a fixed limited sampling budget, a slower sampling

process restricts the number of candidate images produced, narrowing the selection range, and thus limiting the achievable image quality. Moreover, selection based on a one-time evaluation over a finite sample pool typically yields merely identifying the least flawed result rather than a truly high-fidelity image.

To address the above challenges, we propose a novel inference time scaling framework **SF2** aiming to efficiently generate a larger pool of candidate samples within the limited inference budget. Meanwhile, we further propose a reflection mechanism capable of rectifying flaws and refining details of the selected samples during the continue generation process. Specifically, we propose Co-Fusion pipeline that utilizes the blur and low-quality images at the early stage with a fast detail refiner to ensure the visual maintenance of the final output, enabling the estimators to make decisions at the early stage. Also, after performing the selection in the early stage, we leverage the powerful vision language model, *e.g.*, Qwen2.5-VL Bai et al. (2025), to provide the feedback, and inject the feedback through the continue image generation process to rectify defects and enhance fine details of the selected images. To validate the effectiveness of our proposed method, we conduct numerous experiments on current state-of-the-art diffusion models Stable-Diffusion-3.5-Large Esser et al. (2024) and FLUX.1-dev Labs (2024). The experimental results demonstrate that our proposed method can achieve the comparable performance with the same candidate size while accelerate the whole sampling process $2.2 \times$ for Stable-Diffusion-3.5-Large and $2.0 \times$ for FLUX.1-dev. Also, with limited extra inference time costs, our proposed method improves the total quality of Stable-Diffusion-3.5-Large and FLUX.1-dev by 10.84% and 9.90%.

In summary, our contributions can be concluded as follows:

- We propose a novel inference time scaling framework for fast sampling. It can efficiently generate a larger pool of candidate samples via Co-Fusion pipeline within the limited budget while maintaining the main content of the images compared with the original pipeline, providing a wider search space for samples and improve the overall performance;

- We propose a reflection mechanism enables the refinement of details and the rectify the anomalous regions of the selected samples. It injects the feedback of the vision language model into the continue generation process, suppressing undesired or unexpected elements in the generated results;

- Extensive experiments on Stable-Diffusion-3.5-Large and FLUX.1-dev demonstrate the effectiveness of our proposed method. Our proposed method can achieve the comparable performance with the same number of candidates while accelerating the whole sampling process. Also, with limited extra inference time costs, our proposed method improves the total quality of Stable-Diffusion-3.5-Large and FLUX.1-dev by 10.84% and 9.90%.

## 2 RELATED WORKS

### 2.1 INFERENCE TIME SCALING OF DIFFUSION MODELS

Diffusion models Ho et al. (2020); Song et al. (2020); Rombach et al. (2022); Esser et al. (2024); Labs (2024); Podell et al. (2023) excel in text-to-image generation but suffer from randomness and inconsistent quality Zhou et al. (2024); Ahn et al. (2024); Qi et al. (2024); Bai et al. (2024); Xu et al. (2023b). Prior works mainly enhance quality via training-time scaling, *e.g.*, larger models, datasets, or training duration, Mei et al. (2024); Li et al. (2024); Liang et al. (2024); Podell et al. (2023); Chen et al. (2025), which is computationally costly. Inspired by large language models Brown et al. (2024); Snell et al. (2024); Wu et al. (2024a); Gandhi et al. (2024); Balachandran et al. (2025); Zhang et al. (2025), inference time scaling has recently attracted interest in vision Ma et al. (2025); Xie et al. (2025); Kim et al. (2025). Inference-time methods fall into two categories: global trajectory search, finding better initial noise, *e.g.*, random search or Zero-Order Search Ma et al. (2025); and stepwise trajectory search, optimizing update directions during denoising, e.g., Search over Paths Ma et al. (2025). The two are orthogonal, targeting the starting point and the generation process, and can be combined. Our method focuses on efficiency and can serve as a plug-and-play enhancement.

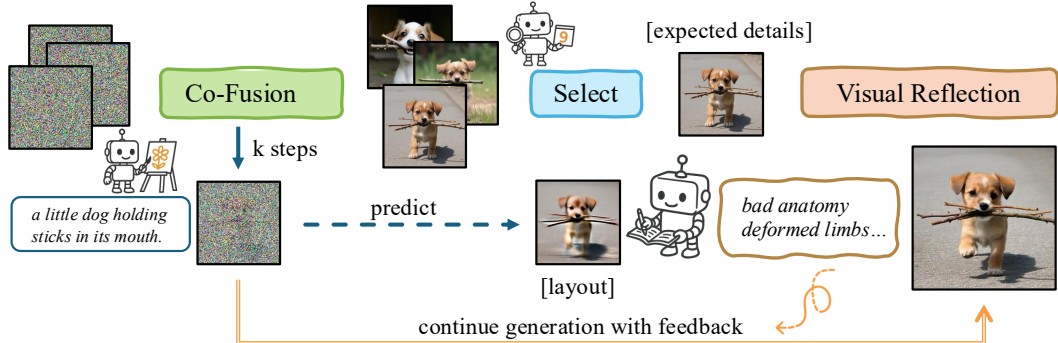

Figure 1: The overall framework of our proposed method. Our proposed method includes three parts. Firstly, we introduce Co-Fusion pipeline to fast sample larger set of candidates. Then, we adopt pre-defined evaluation metrics to select the highest fidelity candidate within the set. Lastly, we leverage vision language model to provide feedback of the potential flaws in the reference images, and inject the feedback to the continue generation process from early stage.

## 2.2 ALIGNMENT OF DIFFUSION MODELS

Alignment Liu et al. (2024); Lee et al. (2023); Xu et al. (2023a); Black et al. (2023); Fan et al. (2023); Clark et al. (2023); Wallace et al. (2024); Yang et al. (2024c;a); Liang (2024) adapts pretrained diffusion models for human-preferred outputs beyond training distribution. The two main training-based strategies are RLHF (learning a reward model then finetuning) Lee et al. (2023); Xu et al. (2023a); Black et al. (2023); Fan et al. (2023); Clark et al. (2023); Zhang et al. (2024); Uehara et al. (2024) and DPO (direct optimization on preference data without reward models) Wallace et al. (2024); Yang et al. (2024c;a); Liang (2024); Yuan et al. (2024); Zhu et al. (2025); Na et al. (2024). Other alignment methods avoid finetuning, such as prompt optimization Hao et al. (2023); Wang et al. (2023); Mañas et al. (2024); Mo et al. (2024), noise optimization Eyring et al. (2024); Tang et al. (2024); Bai et al. (2024); Zhou et al. (2024); Qi et al. (2024), and attention control Wu et al. (2024b); Yang et al. (2024b); Hong et al. (2023). Alignment refines the objective of models to favor human preferences, while inference time scaling searches within the output space to exploit model capacity. These two approaches are complementary, and our method can be integrated with both to further improve image quality.

## 3 METHOD

### 3.1 PRELIMINARY

Assume the time cost of full sampling process is $\tau$, and the total time budget for inference time scaling is $\mathcal{T}$, the number of candidate samples can be generated is $n = \lfloor \frac{\mathcal{T}}{\tau} \rfloor$. For each $i = 1, \ldots, n$, $x_0^{(i)} \sim \mathcal{N}(0, I)$, and $I^{(i)} = G(x^{(i)}, \theta)$. Here, $G(\cdot; \theta)$ refers the total denoising process mapping parameterized by $\theta$. After repeated sampling process, the final images are selected from the candidate sets with the predefined metric $R(\cdot)$,

$$I^* = \underset{I^{(I)} \in \{I^{(0)}, \ldots, I^{(n)}\}}{\arg\max} R(I^{(i)}). \tag{1}$$

Denote cumulative distribution function of each score $R^{(i)} = R(I^{(i)})$ as $F_R(\gamma) = P(R^{(i)} \leq \gamma)$, the expected maximum score over $n$ samples is as follows:

$$Q(n) = \mathbb{E}\left[\max_{1 \leq i \leq n} R^{(i)}\right] = \int_{-\infty}^{\infty} P(\max_i R^{(i)} > \gamma)\,\mathrm{d}\gamma = \int_{-\infty}^{\infty} \left[1 - F_R(\gamma)^n\right]\,\mathrm{d}\gamma. \tag{2}$$

As $n$ increases, the distribution shifts rightward, yielding higher expected scores, but marginal gains diminish accordingly.

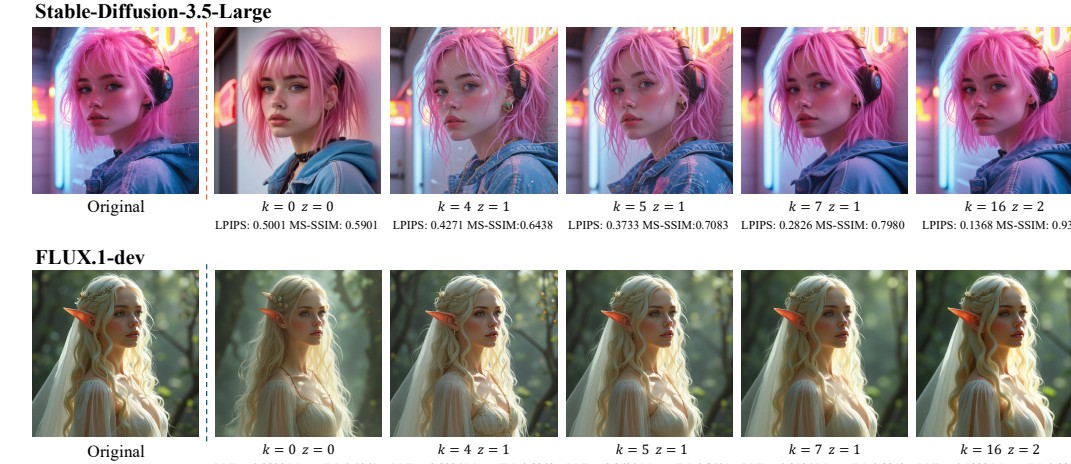

Figure 2: The visualization and quantitative results for different choice of $k$ and $z$. The results demonstrate that smaller $\Delta$ contributes to more similar production.

## 3.2 CO-FUSION PIPELINE

Current widely-adopted inference time scaling method requires repeated sampling through the whole pipeline, which is time-consuming. Also, as presented in Eq. 2, the increase in number of candidate samples benefits the overall expected score. In this case, we propose Co-Fusion pipeline. It accelerates the whole sampling process, while maintaining the visual of the outputs, so that the estimators can make decisions in the early steps by consensus.

Specifically, Co-Fusion pipeline involves two stages. First, we adopt the original model performing $k$ denoising steps to capture coarse layout of the images. Then, the generation is switched to the distilled Turbo model to quickly and seamlessly refine the details. The visual results of Co-Fusion are maintained, as the generation process is constrained under the capability of the original model. Here, assume the original model with parameter $\theta$ requires $T$ denoising steps for generation, and the timestep is denoted as $t_i$ for step $i$; the distilled Turbo model $\psi$ only requires $S$ denoising steps, and $S << T$, and the corresponding timestep is denoted as $s_i$ for step $i$. Co-Fusion can be formulated as follows:

$$x_{\theta,t_{i+1}} = x_{\theta,t_i} + \delta_{t_{i+1}} v_\theta(x_{\theta,t_i}, y; t_i), \ where \ i = 0, 1, \ldots, k-1, \ x_{\theta,t_0} \sim \mathcal{N}(0, I),$$
$$x_{\psi,s_{j+1}} = x_{\psi,s_j} + \delta_{s_{j+1}} v_\psi(x_{\psi,s_j}, y; s_j), \ where \ j = z, \ldots, S-1, \ x_{\psi,s_z} = x_{\theta,t_k}, \tag{3}$$

where $\delta_{t_{i+1}} = t_{i+1} - t_i$, and $\delta_{t_{j+1}} = t_{j+1} - t_j$. $x_{\psi,s_S} = x_N$ is the final output of the Co-Fusion pipeline, and the number of the total steps is $N = k + S - z < T$. $y$ refer to the condition for image generation, *e.g.*, text prompt. We also provide theoretical analysis of error bound of Co-Fusion pipeline to further validate its effectiveness. For more detailed theoretical analysis, please refer to the supplementary.

**Proposition 1.** *Error bound for Co-Fusion pipeline*

*Proof.* For the first $k$ steps, Co-Fusion and the original pipeline share the same process. Thus, in this case, $e_{\theta,0\to k} = 0$, and $x_{\theta,t_k} = x_{\psi,s_z}$. For the rest original pipeline $T - k$ steps, we have:

$$e_{\theta,i} = x_{\theta,t_i} - x^*_{\theta,t_i}, \ \varepsilon_{\theta,i} = x^*_{\theta,t_i} - x^*_{\theta,t_{i-1}} - \delta_{t_i} v_\theta(x^*_{\theta,t_{i-1}}, y; t_{i-1}), \ ||\varepsilon_{\theta,i}|| \le C\delta_{t_i}^2,$$
$$e_{\theta,i} = e_{\theta,i-1} + \delta_{t_i}[v_\theta(x_{\theta,t_{i-1}}, y; t_{i-1}) - v_\theta(x^*_{\theta,t_{i-1}}, y; t_{i-1})] - \varepsilon_{\theta,i}. \tag{4}$$

Here, denote the Lipschitz constant for $v_\theta$ and $x$ as $L$, we have:

$$||v_\theta(x_{\theta,t_{i-1}}, y; t_{i-1}) - v_\theta(x^*_{\theta,t_{i-1}}, y; t_{i-1})|| \le L||x_{\theta,t_{i-1}} - x^*_{\theta,t_{i-1}}|| = L||e_{\theta,i-1}||,$$
$$||e_{\theta,i}|| \le ||e_{\theta,i-1}|| + ||\delta_{t_i}[v_\theta(x_{\theta,t_{i-1}}, y; t_{i-1}) - v_\theta(x^*_{\theta,t_{i-1}}, y; t_{i-1})]|| + ||\varepsilon_{\theta,i}|| \tag{5}$$
$$\le (1 + \delta_{t_i}L)||e_{\theta,i-1}|| + C\delta_{t_i}^2.$$

After iterative expansion, we have:

$$||\boldsymbol{e}_{\theta,T}|| \leq \left[\prod_{m=k+1}^{T} (1+\delta_{t_m}L)\right] ||\boldsymbol{e}_{\theta,k}|| + C \sum_{m=k+1}^{T} \delta_{t_m}^2 \prod_{p=m+1}^{T} (1+\delta_{t_p}L),$$

$$1+\delta L \leq e^{\delta L}, \ ||\boldsymbol{e}_{\theta,T}|| \leq e^{(t_T-t_k)L}||\boldsymbol{e}_{\theta,k}|| + C \sum_{m=k+1}^{T} \delta_{t_m}^2 e^{(t_T-t_m)L}. \tag{6}$$

Denote $\Delta_{ori} = t_T - t_k$, $\delta_{max_t} = \max\limits_{i} \delta_{t_i}$, $\sum_{m=k+1}^{T} \delta_{t_m}^2 \leq \Delta_{ori}\delta_{max_t}$, $e^{\Delta_{ori}L}\Delta_{ori} \leq \frac{e^{\Delta_{ori}L}-1}{L}$, then we have:

$$||\boldsymbol{e}_{\theta,T}|| \leq e^{\Delta_{ori}L}||\boldsymbol{e}_{\theta,k}|| + \frac{e^{\Delta_{ori}L}-1}{L}C\delta_{max_t}. \tag{7}$$

For the rest Co-Fusion $S - z$ steps, we have:

$$x_{\psi,j} = x_{\psi,j-1} + \delta_{s_j}v_{\psi}(x_{\psi,j-1}, y; s_{j-1}) = x_{\psi,j-1} + \delta_{s_j}[v_{\theta}(x_{\psi,j-1}, y; s_{j-1}) + \varsigma_{s_{j-1}}],$$

$$\boldsymbol{e}_{\psi,j} = x_{\psi,s_j} - x^*_{\theta,s_j}, \ \varepsilon_{\theta,j} = x^*_{\theta,s_j} - x^*_{\theta,s_{j-1}} - \delta_{s_j}v_{\theta}(x^*_{\theta,s_{j-1}}; s_{j-1}), \ ||\varepsilon_{\theta,j}|| \leq C\delta_{s_j}^2, \tag{8}$$

$$\boldsymbol{e}_{\psi,j} = \boldsymbol{e}_{\psi,j-1} + \delta_{s_j}[v_{\theta}(x_{\psi,s_{j-1}}, y; s_{j-1}) - v_{\theta}(x^*_{\psi,s_{j-1}}, y; s_{j-1})] + \delta_{s_j}\varsigma_{s_{j-1}} - \varepsilon_{\theta,j}.$$

Here, denote the distillation error bound as the constant $\varepsilon_{dis}$, $||\varsigma|| \leq \varepsilon_{dis}$, and the Lipschitz constant for $v_{\theta}$ and $x$ as $L$, we have:

$$||\boldsymbol{e}_{\psi,j}|| \leq (1+\delta_{s_j}L)||\boldsymbol{e}_{\psi,j-1}|| + C\delta_{s_j}^2 + \delta_{s_j}||\varsigma_{s_{j-1}}||$$

$$\leq (1+\delta_{s_j}L)||\boldsymbol{e}_{\psi,j-1}|| + C\delta_{s_j}^2 + \delta_{s_j}\varepsilon_{dis} \tag{9}$$

Similarly, we apply iterative expansion to Eq. 9, and we have:

$$||\boldsymbol{e}_{\psi,S}|| \leq \left[\prod_{m=z+1}^{S} (1+\delta_{s_m}L)\right] ||\boldsymbol{e}_{\psi,z}|| + \sum_{m=z+1}^{S} (C\delta_{s_m}^2 + \delta_{s_m}\varepsilon_{dis}) \prod_{p=m+1}^{S} (1+\delta_{s_m}L)$$

$$\leq e^{\Delta_{co}L}||\boldsymbol{e}_{\psi,z}|| + \frac{e^{\Delta_{co}L}-1}{L}(C\delta_{max_s} + \varepsilon_{dis}), \tag{10}$$

where $\Delta_{co} = s_S - s_z = \Delta_{ori}$, we denote it as $\Delta$ for clarification, and $||\boldsymbol{e}_{\psi,z}|| = ||\boldsymbol{e}_{\theta,k}|| = 0$, $\delta_{max_s} = \max\limits_{j} \delta_{s_j}$. The final error bound for Co-Fusion pipeline is as follows:

$$||x_{\theta,t_T} - x_{\psi,s_S}|| \leq \frac{e^{\Delta \cdot L}-1}{L}[C(\delta_{max_t} + \delta_{max_s}) + \varepsilon_{dis}]. \tag{11}$$

The error for the Co-Fusion pipeline is bounded with the length of the left steps and the distillation loss between the two models. □

### 3.3 VISUAL REFLECTION

After selection part performed in the early stage, the selected trajectory will continue to generate the final output with the original model. However, as mentioned in Eq. 2, the performance of the final output is bounded by the size of the candidate set. Meanwhile, from the observation, the selected images are the least imperfect among the candidates and can not guarantee the high-fidelity. To further improve the quality of the selected images, we propose the visual reflection method during the continue generation process.

Specifically, we first project the noisy image at step $k$ into the predicted final result:

$$\hat{x}_{\theta,t_T} = x_{\theta,t_k} - t_k v_{\theta}(x_{\theta,t_k}, y; t_k), \tag{12}$$

which presents the layout of the image. Then, we leverage the visual understanding capabilities of the vision language model $\mathcal{M}$, *e.g.*, Qwen2.5-VL, to point out the flaws of the reference images (the predicted image $\hat{x}_{\theta,t_T}$ and the corresponding candidate image $x_N$ generated through Co-Fusion pipeline), such as distorted figures, low fidelity, *etc*. Here, the predicted image can reflect the overall

---

**Algorithm 1:** SF2 Framework

---

**Input:** Original model with parameter $\theta$, turbo model with parameter $\psi$, vision language model $\mathcal{M}$, evaluation metric $R$, prompt $y$, the number of candidates $n$

**Output:** High-fidelity $x'_{\theta,T}$ for prompt $y$

Random initialize $x_{\theta,t_0} \sim \mathcal{N}(0, I)$

\# Candidates generation

**for** $idx = 1$ *to* $n$ **do**

    \# Co-Fusion pipeline

    Generate candidate $x_N^{(idx)}$ through Eq. 3          ▷ Store $k$ step $x_{\theta,t_k}^{(idx)}$ for continue generation

    $I_N^{(idx)} = \mathcal{D}_\psi \left( x_N^{(idx)} \right)$

$idx^* = \arg\max_{1 \leq idx \leq n} R\left( I^{(idx)} \right)$

\# Visual reflection

Get predicted final output $\hat{x}_{\theta,t_T}^{(idx^*)}$ with $x_{\theta,t_k}^{(idx^*)}$ through Eq. 12

Generate feedback of the flaws in the reference images through vision language model $\mathcal{M}$,

$y' = \mathcal{M}\left( \hat{I}_{\theta,t_T}^{(idx^*)}, I_N^{(idx^*)}, p \right)$, where $\hat{I}_{\theta,t_T}^{(idx^*)} = \mathcal{D}_\theta \left( \hat{x}_{\theta,t_T}^{(idx^*)} \right)$

\# Continue generation

Generate the final result $x'_{\theta,T}$ with $x_{\theta,t_k}^{(idx^*)}$ through Eq. 14.

**return** $x'_{\theta,T}$

---

layout of the final image and the candidate image can demonstrate the expected details of the final image. The process can be formulated as follows:

$$\hat{I}_{\theta,t_T} = \mathcal{D}_\theta(\hat{x}_{\theta,t_T}), \ I_N = \mathcal{D}_\psi(x_N)$$
$$y' = \mathcal{M}(\hat{I}_{\theta,t_T}, I_N, p), \tag{13}$$

where $y'$ is the negative prompt identifying the potential defects of the selected images, $\mathcal{D}$ is decoder, $p$ refer to the instruction prompt. After that, we inject the feedback $y'$ into the continue generation process from the breakpoint $k$ to get the final output $x'_{\theta,T}$:

$$x'_{\theta,t_{i+1}} = x_{\theta,t_i} + \delta_{t_{i+1}} v_\theta(x_{\theta,t_i}, y, y'; t_i), \ where \ i = k, \ldots, T-1. \tag{14}$$

Since the early step $k$ is still in layout generation phase, it exhibits effective corrective efficacy for mitigating potential flaws, while preserving the initial favorable layout and the trajectory of fine-detail synthesis.

### 3.4 Algorithm Summary

In this paper, we propose a novel inference time scaling framework for fast and broader sampling with early trajectory selection and reflection. The algorithm summary is shown in Algorithm 1. Our proposed framework includes three parts. Firstly, we fast build the candidate set through our proposed Co-Fusion pipeline, which enables the estimator to make decision in the early stage. After selection, we leverage the vision language model to generate feedback on the flaws of the reference images (the predicted image $\hat{I}_{\theta,t_T}^{(idx^*)}$ in the $k$ step and the selected candidate image $I_N^{(idx^*)}$ generated through Co-Fusion pipeline). These reference images can demonstrate the layout and expected details information, respectively, offering a broader view for vision language model to provide correct and effective feedback. Then, we inject the generated feedback into the continue generation process from the breakpoint $k$ of the original pipeline to get the final output $x'_{\theta,T}$. As $k$ is in early stage of the whole pipeline, our method can smoothly correct the potential flaws while preserving the high-quality layout and preventing artifact generation.

Table 1: Quantitative results of baseline methods and our proposed method. Here, NFEs records the total budget each prompt requires to obtain the final image. Avg RI denotes the average relative improvement of the four evaluation metrics over the original.

| Method/Settings | NFEs | Estimator | HPSv2.1 | ImageReward | PickScore | Aesthetic Score | Avg RI (%) |
|---|---|---|---|---|---|---|---|
| **Stable-Diffusion-3.5-Large** | | | | | | | |
| Original | 28 | - | 30.95 | 1.0441 | 23.03 | 5.435 | - |
| SDS | 280 | - | 30.31 | 1.1220 | 22.73 | 5.414 | 0.93% |
| Random ($n = 10$) | 280 | - | 30.80 | 1.0408 | 23.02 | 5.435 | -0.21% |
| BoN ($n = 10$) | 280 | HPSv2.1 | 32.78 | 1.2095 | 23.37 | 5.493 | 6.07% |
| **Ours** ($n = 10$) | **150** | HPSv2.1 | 32.92 | 1.2164 | 23.38 | 5.595 | 6.83% |
| **Ours** ($n = 20$) | **280** | HPSv2.1 | 33.22 | 1.2418 | 23.43 | 5.609 | 7.80% |
| BoN ($n = 10$) | 280 | ImageReward | 31.64 | 1.4058 | 23.22 | 5.464 | 9.56% |
| **Ours** ($n = 10$) | **150** | ImageReward | 31.98 | 1.3731 | 23.26 | 5.568 | 9.57% |
| **Ours** ($n = 20$) | **280** | ImageReward | 32.10 | 1.4184 | 23.29 | 5.579 | 10.84% |
| BoN ($n = 10$) | 280 | PickScore | 31.88 | 1.1751 | 23.67 | 5.490 | 4.84% |
| **Ours** ($n = 10$) | **150** | PickScore | 32.02 | 1.1893 | 23.59 | 5.590 | 5.66% |
| **Ours** ($n = 20$) | **280** | PickScore | 32.31 | 1.2133 | 23.70 | 5.604 | 6.65% |
| **FLUX.1-dev** | | | | | | | |
| Original | 28 | - | 32.08 | 1.0835 | 23.32 | 5.719 | - |
| SDS | 280 | - | 31.69 | 1.0835 | 23.19 | 5.680 | -0.61% |
| Random ($n = 10$) | 280 | - | 32.00 | 1.0720 | 23.30 | 5.702 | -0.42% |
| BoN ($n = 10$) | 280 | HPSv2.1 | 33.73 | 1.2334 | 23.66 | 5.744 | 5.22% |
| **Ours** ($n = 10$) | **130** | HPSv2.1 | 33.64 | 1.2392 | 23.64 | 5.805 | 5.53% |
| **Ours** ($n = 20$) | **240** | HPSv2.1 | 33.94 | 1.2586 | 23.70 | 5.817 | 6.33% |
| BoN ($n = 10$) | 280 | ImageReward | 32.61 | 1.4441 | 23.50 | 5.715 | 8.91% |
| **Ours** ($n = 10$) | **130** | ImageReward | 32.70 | 1.4163 | 23.50 | 5.788 | 8.66% |
| **Ours** ($n = 20$) | **240** | ImageReward | 32.82 | 1.4627 | 23.54 | 5.796 | 9.90% |
| BoN ($n = 10$) | 280 | PickScore | 32.91 | 1.2069 | 23.96 | 5.738 | 4.26% |
| **Ours** ($n = 10$) | **130** | PickScore | 32.93 | 1.2180 | 23.88 | 5.804 | 4.74% |
| **Ours** ($n = 20$) | **240** | PickScore | 33.07 | 1.2340 | 23.99 | 5.810 | 5.36% |

Table 2: Total runtime for each prompt to obtain the final image via different inference time scaling method. Here, the runtime includes image generation and selection parts. For our proposed method, it includes candidate image generation through Co-Fusion pipeline, image selection, visual reflection and continue generation. Here, we report the total runtime with HPSv2.1, ImageReward, and PickScore estimators, and the results are separated by the slash.

| | | | **Stable-Diffusion-3.5-Large** | | |
|---|---|---|---|---|---|
| Original | SDS | Random ($n = 10$) | BoN ($n = 10$) | **Ours** ($n = 10$) | **Ours** ($n = 20$) |
| 19.82s | 195.28s | 197.12s | 204.60s / 198.19s / 198.12s | **98.55s / 92.15s / 92.08s** | **182.87s / 170.40s / 170.38s** |
| | | | **FLUX.1-dev** | | |
| Original | SDS | Random ($n = 10$) | BoN ($n = 10$) | **Ours** ($n = 10$) | **Ours** ($n = 20$) |
| 22.50s | 223.04s | 225.54s | 233.02s / 226.61s / 226.54s | **120.43s / 114.02s / 113.95s** | **221.77s / 209.36s / 209.34s** |

## 4 EXPERIMENTS

### 4.1 SETTINGS AND IMPLEMENTATION DETAILS

We conduct experiments on the open-source diffusion models Stable-Diffusion-3.5-Large and FLUX.1-dev to validate our proposed method SF2. For each prompt, we set 280 NFEs as the inference budget. The default number of generation steps is 28 (28 NFEs), for both two models across all experimental settings. For the original model and scale denoising steps (SDS) method, the seed is 42. For the rest methods (random select, best-of-n (BoN), and ours), the seeds are randomly sampled from a uniform distribution (the value is $0 \sim 10,000$), but remain same for the three methods. For our proposed method, we adopt the corresponding Turbo models (SD-3.5-Large-Turbo and

**Stable-Diffusion-3.5-Large**

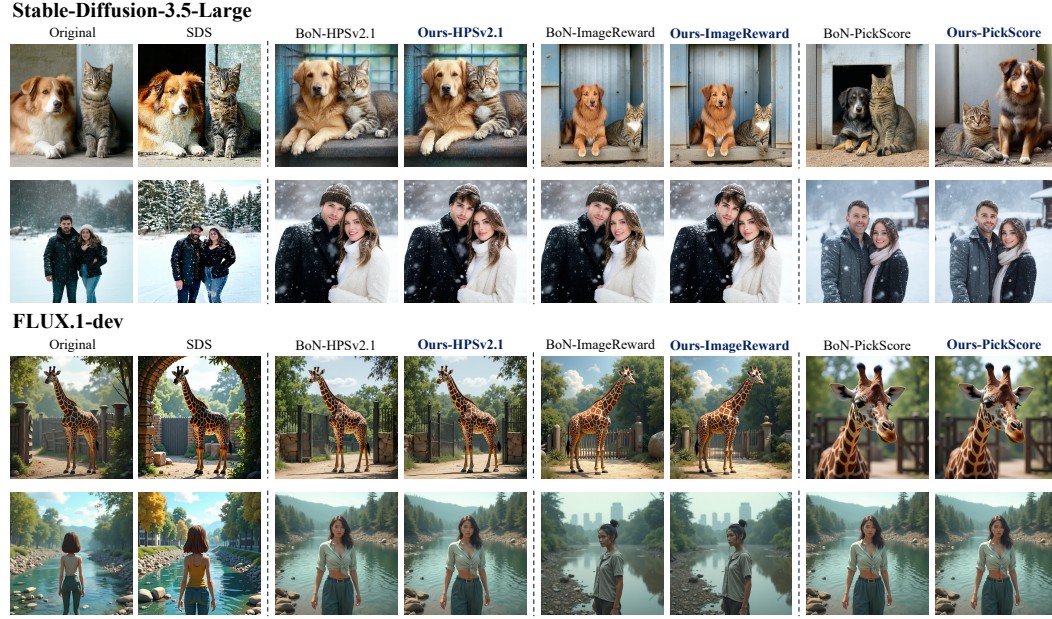

Figure 3: The visualization of our proposed method and BoN. The candidate images are generated with the prompts from COCO2014val dataset under the setting of $n = 10$.

FLUX.1-dev-Turbo) to collaboratively perform the Co-Fusion pipeline. The default number of steps for generation is $8$ and $4$ for SD-3.5-Large-Turbo and FLUX.1-dev respectively. Specifically, we set $k = 7$ and $z = 3$ for SD-3.5-Large and $k = 7$ and $z = 1$ for FLUX.1-dev. For visual reflection process, we adopt Qwen2.5-VL-7B model to process the selected images and generate feedback. For dataset, we adopt $5,000$ prompts randomly selected from COCO2014val dataset Lin et al. (2014), and generate $n$ candidates for each prompt. We adopt HPSv2.1 Wu et al. (2023), ImageReward Xu et al. (2023a), and PickScore Kirstain et al. (2023) as the estimator, and the performance are evaluated on the HPSv2.1, ImageReward, PickScore, and Aesthetic Score. We also report the required NFEs and runtime (s) to further demonstrate the efficiency of our proposed method. All experiments are conducted on NVIDIA RTX 6000 Ada. For more details and results, please refer to Appendix.

### 4.2 MAIN RESULTS

We evaluate our method on $5,000$ prompts from COCO2014val dataset. For each prompt, the limited budget for generation and search is 280 NFEs. The results are shown in Table 1. We also evaluate the total runtime of the inference time scaling process for each prompt as shown in Table 2. The total runtime includes image generation and selection parts, and our method includes candidate image generation through Co-Fusion pipeline, image selection, visual reflection and continue generation. The visualization results are presented in Fig. 3. The results show that our proposed method can achieve comparable performance while only requires 53.57% and 46.43% NFEs budget of BoN. With similar NFEs budget, our proposed method substantially outperforms BoN method. For instance, our method improves FLUX.1-dev model by 9.90% with 240 NFEs compared with BoN method 8.91% with 280 NFEs. Moreover, the visualization results also show that our method can effectively restore the anomalous regions in the images, and enhance overall quality of images.

### 4.3 ABLATION STUDIES

We also conduct ablation studies to validate the effectiveness of each component proposed in our proposed method. The experimental settings are same with the baseline comparison. The results are demonstrated in Table 3. From the results, our proposed Co-Fusion pipeline can quickly and accurately restore the outputs of the original pipeline, enabling precise selection while significantly reducing required extra time. Moreover, our proposed visual reflection effectively corrects unnatural

Table 3: Ablation studies for each component of our proposed method. NFEs reports the total required budget for each prompt to obtain the final image. Avg RI records the average relative improvement of the four evaluation metrics over the original.

| Method/Settings | NFEs | Estimator | HPSv2.1 | ImageReward | PickScore | Aesthetic Score | Avg RI (%) |
|---|---|---|---|---|---|---|---|
| **Stable-Diffusion-3.5-Large** | | | | | | | |
| Original | 28 | - | 30.95 | 1.0441 | 23.03 | 5.435 | - |
| + Co-Fusion ($n = 10$) | 150 | HPSv2.1 | 32.61 | 1.2014 | 23.34 | 5.492 | 5.71% |
| + Visual-Reflection ($n = 10$) | 150 | HPSv2.1 | 32.92 | 1.2164 | 23.38 | 5.595 | 6.83% |
| + Co-Fusion ($n = 20$) | 280 | HPSv2.1 | 32.92 | 1.2267 | 23.41 | 5.496 | 6.66% |
| + Visual-Reflection ($n = 20$) | 280 | HPSv2.1 | 33.22 | 1.2418 | 23.43 | 5.609 | 7.80% |
| + Co-Fusion ($n = 10$) | 150 | ImageReward | 31.62 | 1.3662 | 23.22 | 5.463 | 8.59% |
| + Visual-Reflection ($n = 10$) | 150 | ImageReward | 31.98 | 1.3731 | 23.26 | 5.568 | 9.57% |
| + Co-Fusion ($n = 20$) | 280 | ImageReward | 31.74 | 1.4137 | 23.25 | 5.460 | 9.84% |
| + Visual-Reflection ($n = 20$) | 280 | ImageReward | 32.10 | 1.4184 | 23.29 | 5.579 | 10.84% |
| + Co-Fusion ($n = 10$) | 150 | PickScore | 31.78 | 1.1719 | 23.57 | 5.490 | 4.57% |
| + Visual-Reflection ($n = 10$) | 150 | PickScore | 32.02 | 1.1893 | 23.59 | 5.590 | 5.66% |
| + Co-Fusion ($n = 20$) | 280 | PickScore | 31.97 | 1.1946 | 23.68 | 5.494 | 5.40% |
| + Visual-Reflection ($n = 20$) | 280 | PickScore | 32.31 | 1.2133 | 23.70 | 5.604 | 6.65% |
| **FLUX.1-dev** | | | | | | | |
| Original | 28 | - | 32.08 | 1.0835 | 23.32 | 5.719 | - |
| + Co-Fusion ($n = 10$) | 130 | HPSv2.1 | 33.59 | 1.2319 | 23.63 | 5.740 | 5.02% |
| + Visual-Reflection ($n = 10$) | 130 | HPSv2.1 | 33.64 | 1.2392 | 23.64 | 5.805 | 5.53% |
| + Co-Fusion ($n = 20$) | 240 | HPSv2.1 | 33.89 | 1.2534 | 23.69 | 5.749 | 5.86% |
| + Visual-Reflection ($n = 20$) | 240 | HPSv2.1 | 33.94 | 1.2586 | 23.70 | 5.817 | 6.33% |
| + Co-Fusion ($n = 10$) | 130 | ImageReward | 32.62 | 1.4115 | 23.49 | 5.712 | 8.14% |
| + Visual-Reflection ($n = 10$) | 130 | ImageReward | 32.70 | 1.4163 | 23.50 | 5.788 | 8.66% |
| + Co-Fusion ($n = 20$) | 240 | ImageReward | 32.73 | 1.4595 | 23.52 | 5.721 | 9.41% |
| + Visual-Reflection ($n = 20$) | 240 | ImageReward | 32.82 | 1.4627 | 23.54 | 5.796 | 9.90% |
| + Co-Fusion ($n = 10$) | 130 | PickScore | 32.87 | 1.2076 | 23.88 | 5.736 | 4.15% |
| + Visual-Reflection ($n = 10$) | 130 | PickScore | 32.93 | 1.2180 | 23.88 | 5.804 | 4.74% |
| + Co-Fusion ($n = 20$) | 240 | PickScore | 33.01 | 1.2262 | 23.99 | 5.744 | 4.84% |
| + Visual-Reflection ($n = 20$) | 240 | PickScore | 33.07 | 1.2340 | 23.99 | 5.810 | 5.36% |

artifacts in the source images and enhances the overall image quality, while preserving most of the original content, *e.g.*, layout, fine details. It can also be evidenced by the visualization results in Fig. 3. Also, the results demonstrate that increasing $n$ can improve the quality of the final results.

## 4.4 DISCUSSION

The experimental results indicate that the final quality of the selected image is highly correlated with the adopted estimator. Although the estimators HPSv2.1, ImageReward, and PickScore are all trained on user preference datasets, which incorporate both aesthetics and prompt following, they diverge in their selection due to unavoidable biases in training data. Nevertheless, these models remain interrelated: using any estimator also boosts performance on the other metrics. Our proposed method effectively balances all four evaluation metrics by further incorporating visual reflection to restore the unnatural artifacts and improve the quality of the final image.

## 5 CONCLUSION

In this paper, we propose a novel efficient inference time scaling framework **SF2** for visual generation. First, we propose Co-Fusion pipeline for fast and broader sampling, while maintaining the similar image content for estimators to make decision. It can efficiently generate a larger candidate pool within the limited budget, providing a wider search space and improving the overall achievable performance. After selection, we propose visual reflection mechanism to rectify the unnatural regions of the selected images and further refine the overall details, while preserving the main contents. Extensive experiments demonstrate the effectiveness of our proposed method.

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

# A  APPENDIX

## A.1  THEORETICAL ANALYSIS FOR ERROR BOUND OF CO-FUSION PIPELINE

Co-Fusion pipeline has total inference steps $N = k + S - z < T$. Here, $S$ refers to the number of total inference steps for Turbo model, and $T$ refers to the number of total inference steps for the original model. For the first $k$ steps, Co-Fusion and the original pipeline share the same process. Thus, in this case, the error for the first $k$ steps $e_{\theta,0 \to k} = 0$, and $x_{\theta,t_k} = x_{\psi,s_z}$. For the rest original pipeline $T - k$ steps, we have:

$$
e_{\theta,i} = x_{\theta,t_i} - x^*_{\theta,t_i}, \; \varepsilon_{\theta,i} = x^*_{\theta,t_i} - x^*_{\theta,t_{i-1}} - \delta_{t_i} v_\theta(x^*_{\theta,t_{i-1}}, y; t_{i-1}), \; ||\varepsilon_{\theta,i}|| \le C\delta^2_{t_i},
$$
$$
e_{\theta,i} = e_{\theta,i-1} + \delta_{t_i}[v_\theta(x_{\theta,t_{i-1}}, y; t_{i-1}) - v_\theta(x^*_{\theta,t_{i-1}}, y; t_{i-1})] - \varepsilon_{\theta,i}.
$$
(15)

Here, denote the Lipschitz constant for $v_\theta$ and $x$ as $L$, we have:

$$
||v_\theta(x_{\theta,t_{i-1}}, y; t_{i-1}) - v_\theta(x^*_{\theta,t_{i-1}}, y; t_{i-1})|| \le L||x_{\theta,t_{i-1}} - x^*_{\theta,t_{i-1}}|| = L||e_{\theta,i-1}||,
$$
$$
||e_{\theta,i}|| = ||e_{\theta,i-1} + \delta_{t_i}[v_\theta(x_{\theta,t_{i-1}}, y; t_{i-1}) - v_\theta(x^*_{\theta,t_{i-1}}, y; t_{i-1})] - \varepsilon_{\theta,i}||
$$
$$
\le ||e_{\theta,i-1}|| + ||\delta_{t_i}[v_\theta(x_{\theta,t_{i-1}}, y; t_{i-1}) - v_\theta(x^*_{\theta,t_{i-1}}, y; t_{i-1})]|| + ||\varepsilon_{\theta,i}||
$$
$$
= ||e_{\theta,i-1}|| + \delta_{t_i}L||e_{\theta,i-1}|| + ||\varepsilon_{\theta,i}||
$$
$$
\le (1 + \delta_{t_i}L)||e_{\theta,i-1}|| + C\delta^2_{t_i}.
$$
(16)

After iterative expansion, we have:

$$
||e_{\theta,T}|| \le \left[\prod_{m=k+1}^{T}(1 + \delta_{t_m}L)\right]||e_{\theta,k}|| + C\sum_{m=k+1}^{T}\delta^2_{t_m}\prod_{p=m+1}^{T}(1 + \delta_{t_p}L),
$$
$$
1 + \delta L \le e^{\delta L}, \; ||e_{\theta,T}|| \le e^{(t_T - t_k)L}||e_{\theta,k}|| + C\sum_{m=k+1}^{T}\delta^2_{t_m}e^{(t_T - t_m)L}.
$$
(17)

Denote $\Delta_{ori} = t_T - t_k$, $\delta_{max_t} = \max_i \delta_{t_i}$, $\sum_{m=k+1}^{T}\delta^2_{t_m} \le \Delta_{ori}\delta_{max_t}$, $e^{\Delta_{ori}L}\Delta_{ori} \le \frac{e^{\Delta_{ori}L} - 1}{L}$, then we have:

$$
||e_{\theta,T}|| \le e^{\Delta_{ori}L}||e_{\theta,k}|| + \frac{e^{\Delta_{ori}L} - 1}{L}C\delta_{max_t}.
$$
(18)

For the rest Co-Fusion $S - z$ steps, we have:

$$
x_{\psi,j} = x_{\psi,j-1} + \delta_{s_j}v_\psi(x_{\psi,j-1}, y; s_{j-1}) = x_{\psi,j-1} + \delta_{s_j}[v_\theta(x_{\psi,j-1}, y; s_{j-1}) + \varsigma_{s_{j-1}}],
$$
$$
e_{\psi,j} = x_{\psi,s_j} - x^*_{\theta,s_j}, \; \varepsilon_{\theta,j} = x^*_{\theta,s_j} - x^*_{\theta,s_{j-1}} - \delta_{s_j}v_\theta(x^*_{\theta,s_{j-1}}; s_{j-1}), \; ||\varepsilon_{\theta,j}|| \le C\delta^2_{s_j},
$$
$$
e_{\psi,j} = e_{\psi,j-1} + \delta_{s_j}[v_\theta(x_{\psi,s_{j-1}}, y; s_{j-1}) - v_\theta(x^*_{\psi,s_{j-1}}, y; s_{j-1})] + \delta_{s_j}\varsigma_{s_{j-1}} - \varepsilon_{\theta,j}.
$$
(19)

Here, denote the distillation error bound as the constant $\varepsilon_{dis}$, $||\varsigma|| \le \varepsilon_{dis}$, and the Lipschitz constant for $v_\theta$ and $x$ as $L$, we have:

$$
||e_{\psi,j}|| \le (1 + \delta_{s_j}L)||e_{\psi,j-1}|| + C\delta^2_{s_j} + \delta_{s_j}||\varsigma_{s_{j-1}}||
$$
$$
\le (1 + \delta_{s_j}L)||e_{\psi,j-1}|| + C\delta^2_{s_j} + \delta_{s_j}\varepsilon_{dis}
$$
(20)

Table 4: Hyper-parameter settings of Co-Fusion pipeline. Hyper-parameters not listed in table are set as default.

| Models | Num Inference Steps | Guidance Scale | Max Sequence Length | $k$ or $z$ |
|---|---|---|---|---|
| **SD-3.5-Large** | 28 | 3.5 | 256 | $k = 7$ |
| **SD-3.5-large-TurboX** | 8 | 1.5 | 256 | $z = 3$ |
| **FLUX.1-dev** | 28 | 3.5 | 512 | $k = 7$ |
| **FLUX.1-Turbo-Alpha** | 4 | 3.5 | 512 | $z = 1$ |

Similarly, we apply iterative expansion to Eq. 20, and we have:

$$
\begin{aligned}
||\boldsymbol{e}_{\psi,S}|| &\leq \left[ \prod_{m=z+1}^{S} (1 + \delta_{s_m} L) \right] ||e_{\psi,z}|| + \sum_{m=z+1}^{S} (C\delta_{s_m}^2 + \delta_{s_m}\varepsilon_{dis}) \prod_{p=m+1}^{S} (1 + \delta_{s_m} L) \\
&\leq e^{(s_S - s_z)} ||\boldsymbol{e}_{\psi,z}|| + \sum_{m=z+1}^{S} (C\delta_{s_m}^2 + \delta_{s_m}\varepsilon_{dis}) e^{(s_S - s_z)} \qquad\qquad (21) \\
&\leq e^{\Delta_{co} L} ||\boldsymbol{e}_{\psi,z}|| + \frac{e^{\Delta_{co} L} - 1}{L} (C\delta_{max_s} + \varepsilon_{dis}),
\end{aligned}
$$

where $\Delta_{co} = s_S - s_z = \Delta_{ori}$, we denote it as $\Delta$ for clarification, and $||\boldsymbol{e}_{\psi,z}|| = ||\boldsymbol{e}_{\theta,k}|| = 0$, $\delta_{max_s} = \max_j \delta_{s_j}$. The final error bound for Co-Fusion pipeline is as follows:

$$
||x_{\theta,t_T} - x_{\psi,s_S}|| \leq \frac{e^{\Delta \cdot L} - 1}{L} [C(\delta_{max_t} + \delta_{max_s}) + \varepsilon_{dis}]. \qquad\qquad (22)
$$

The error for the Co-Fusion pipeline is bounded with the length of the left steps and the distillation loss between the two models.

### A.2 MORE EXPERIMENTAL DETAILS

In the paper, experiments are conducted on the open-source diffusion models Stable-Diffusion-3.5-Large (*stabilityai/stable-diffusion-3.5-large*) and FLUX.1-dev (*black-forest-labs/FLUX.1-dev*) to validate the effectiveness of our proposed method SF2. For our proposed method, we adopt the corresponding Turbo models Stable-Diffusion-3.5-Large-TurboX (*tensorart/stable-diffusion-3.5-large-TurboX*) and FLUX.1-Turbo-Alpha (*alimama-creative/FLUX.1-Turbo-Alpha*) to collaboratively perform the Co-Fusion pipeline. The default total number of whole denoising process is 28 for Stable-Diffusion-3.5-Large and FLUX.1-dev, and 8 for Stable-Diffusion-3.5-Large-Turbo and 4 FLUX.1-Turbo-Alpha. Specifically, we set $k = 7$ and $z = 3$ for Stable-Diffusiton-3.5-Large and $k = 7$ and $z = 1$ for FLUX.1-dev. For more information about Co-Fusion pipeline hyper-parameter settings, please refer to Table 4.

For dataset, we adopt $5,000$ prompts randomly selected from COCO2014val dataset Lin et al. (2014), and generate $n$ candidates for each prompt. We adopt HPSv2.1 Wu et al. (2023), ImageReward Xu et al. (2023a), and PickScore Kirstain et al. (2023) as the estimator, and the performance are evaluated on the HPSv2.1, ImageReward, PickScore, and Aesthetic Score. We also evaluate the performance on more evaluation metrics, MUSIQ Ke et al. (2021), MAN-IQA Yang et al. (2022), and TOPIQ Chen et al. (2024a), to show the quality of final images. Results are shown in Table 5. For each prompt, we set 280 NFEs as the inference budget. The default number of generation steps is 28, which is 28 NFEs for total denoising process, for both two models across all experimental settings. For the original model and scale denoising steps (SDS) method, the seed is 42. For the rest methods (random select, best-of-n (BoN), and ours), the seeds are randomly sampled from a uniform distribution (the value is $0 \sim 10,000$), but remain same for the three methods.

For visual reflection process, we adopt Qwen2.5-VL-7B (*Qwen/Qwen2.5-VL-7B-Instruct*) model to process the reference images (the selected image generated via Co-Fusion pipeline and the corresponding predicted final image from the early stage) and generate feedback. We also report the required NFEs and runtime (s) to further demonstrate the efficiency of our proposed method. All experiments are conducted on NVIDIA RTX 6000 Ada.

Table 5: Quantitative results of baseline methods and our proposed method. Here, NFEs records the total budget each prompt requires to obtain the final image. Avg RI denotes the average relative improvement of the four evaluation metrics over the original.

| Method/Settings | NFEs | Estimator | MUSIQ ↑ | MAN-IQA ↑ | TOPIQ ↑ | Avg RI (%) |
|---|---|---|---|---|---|---|
| **Stable-Diffusion-3.5-Large** | | | | | | |
| Original | 28 | - | 70.30 | 0.4446 | 0.5527 | - |
| SDS | 280 | - | 65.92 | 0.3314 | 0.4657 | -15.81% |
| Random ($n = 10$) | 280 | - | 70.47 | 0.4566 | 0.5596 | 1.40% |
| BoN ($n = 10$) | 280 | HPSv2.1 | 71.11 | 0.4529 | 0.5644 | 1.71% |
| **Ours** ($n = 10$) | **150** | HPSv2.1 | 72.14 | 0.4650 | 0.5837 | 4.27% |
| **Ours** ($n = 20$) | **280** | HPSv2.1 | 72.28 | 0.4654 | 0.5850 | 4.45% |
| BoN ($n = 10$) | 280 | ImageReward | 70.68 | 0.4520 | 0.5603 | 1.19% |
| **Ours** ($n = 10$) | **150** | ImageReward | 71.79 | 0.4637 | 0.5795 | 3.75% |
| **Ours** ($n = 20$) | **280** | ImageReward | 71.77 | 0.4630 | 0.5799 | 3.72% |
| BoN ($n = 10$) | 280 | PickScore | 70.71 | 0.4537 | 0.5615 | 1.41% |
| **Ours** ($n = 10$) | **150** | PickScore | 71.88 | 0.4656 | 0.5815 | 4.06% |
| **Ours** ($n = 20$) | **280** | PickScore | 71.82 | 0.4639 | 0.5806 | 3.85% |
| **FLUX.1-dev** | | | | | | |
| Original | 28 | - | 70.87 | 0.4642 | 0.6131 | - |
| SDS | 280 | - | 64.79 | 0.3717 | 0.4913 | -16.12% |
| Random ($n = 10$) | 280 | - | 71.27 | 0.4807 | 0.6191 | 1.70% |
| BoN ($n = 10$) | 280 | HPSv2.1 | 71.74 | 0.4817 | 0.6254 | 2.33% |
| **Ours** ($n = 10$) | **130** | HPSv2.1 | 72.22 | 0.4906 | 0.6379 | 3.88% |
| **Ours** ($n = 20$) | **240** | HPSv2.1 | 72.39 | 0.4923 | 0.6396 | 4.17% |
| BoN ($n = 10$) | 280 | ImageReward | 71.40 | 0.4790 | 0.6205 | 1.71% |
| **Ours** ($n = 10$) | **130** | ImageReward | 71.90 | 0.4871 | 0.6321 | 3.16% |
| **Ours** ($n = 20$) | **240** | ImageReward | 71.91 | 0.4883 | 0.6322 | 3.26% |
| BoN ($n = 10$) | 280 | PickScore | 71.23 | 0.4777 | 0.6205 | 1.54% |
| **Ours** ($n = 10$) | **130** | PickScore | 71.61 | 0.4847 | 0.6302 | 2.75% |
| **Ours** ($n = 20$) | **240** | PickScore | 71.63 | 0.4853 | 0.6304 | 2.81% |

## A.3 MORE EXPERIMENTAL RESULTS

### A.3.1 QUANTITATIVE RESULTS ON MORE EVALUATION METRICS

Here, we report more quantitative results with other widely used evaluation metrics, MUSIQ Ke et al. (2021), MAN-IQA Yang et al. (2022), and TOPIQ Chen et al. (2024a), to show the effectiveness of our proposed method. The quantitative results are demonstrated in Table 5. The results demonstrate both the effectiveness and efficiency of our proposed method.

### A.3.2 QUANTITATIVE RESULTS ON OTHER FAST SAMPLERS

Our proposed method supports any type of approximate fast samplers which preserve output similarity to the original pipeline, not necessarily limited to Turbo models. Here, we conduct experiments of applying our proposed framework to DBCache method on FLUX.1-dev model. The prompts are randomly selected from COCO2014val. Here, $k = 7$, and the rest part of Co-Fusion pipeline is carried out by DBCache, and $z = 8$. The estimator we adopt here is HPSv2.1. The results are as shown in the Table 6.

Table 6: Results on other fast sampler. The runtime for Ours+Turbo (in paper) to generate a candidate is 9.45s (compared with original pipeline 22.50s), and the runtime for Ours+DBCache is 14.15s. Evaluations are under $1024 \times 1024$ resolution.

| Model | HPSv2.1 ↑ | ImageReward ↑ | PickScore ↑ |
|---|---|---|---|
| **FLUX.1-dev** | 32.02 | 1.0790 | 23.32 |
| **BoN** | 33.70 | 1.2282 | 23.64 |
| **Ours+Turbo (in paper)** | 33.91 | 1.2513 | 23.69 |
| **Ours+DBCache** | 34.01 | 1.2542 | 23.69 |

Table 7: Results on video generation. Here, the video generation model we adopted is CogVideoX-2B, and the fast sampler is TDM.

| | Original | BoN | Ours w/o Visual Reflection | Ours w/ Visual Reflection |
|---|---|---|---|---|
| **NFEs** | 50 | 300 | 294 | 294 |
| **Visual Quality** ↑ | 0.2339 | 0.4915 | 0.4859 | 0.5264 |

### A.3.3 QUANTITATIVE RESULTS ON VIDEO GENERATION

We also conduct additional experiments applying our reflection mechanism to video modality and high-resolution image generation tasks. The preliminary results demonstrated in Table 7 indicate that our method can effectively generalize to these scenarios, demonstrating promising scalability and adaptability. Here, for video modality, we conduct experiments on CogVideoX-2B model, and the fast generator we adopt is TDM. The total number of inference steps for the original CogVideoX pipeline is 50, and the original TDM pipeline is 4. Here, $k = 25$ and $z = 2$. We randomly select 100 prompts from VBench all_dimension_extended dataset, and for each prompts we generate 6 videos for BoN and 10 videos for our method. Our method can successfully adapt to video modality.

### A.3.4 MORE VISUALIZATION RESULTS

We also present more visualization results in Fig. 4 to demonstrate the effectiveness of our proposed method. Here, all images are of $1024 \times 1024$, and are generated with prompts from COCO2014val dataset.

### A.4 LIMITATIONS

Our proposed method is constrained by the capabilities of the underlying model. In this paper, we focus on enhancing the efficiency and performance of the process of inference time scaling, which is primarily concerned with exploring and leveraging the potential of the model, rather than enhancing the inherent performance of the model. If the base model lacks the ability to generate images that meet the desired requirements, the overall effectiveness of our method will be limited.

### A.5 BROADER IMPACTS

Our proposed method SF2 has positive social impacts. Specifically, we propose a novel inference time scaling method, which enhances the efficiency and performance of the inference time scaling process through the incorporation of Co-Fusion pipeline and visual reflection. Our proposed method offers practial advantages by maximizing the utility of existing pretrained models, thereby reducing the need for resource-intensive retraining or the development of new models. Also, our proposed method avoids redundant sampling process, which saves a lot of computational resources. By enabling more efficient and effective use of pretrained models, our proposed method supports the deployment of visual generation models in various real-world applications.

**Stable-Diffusion-3.5-Large**

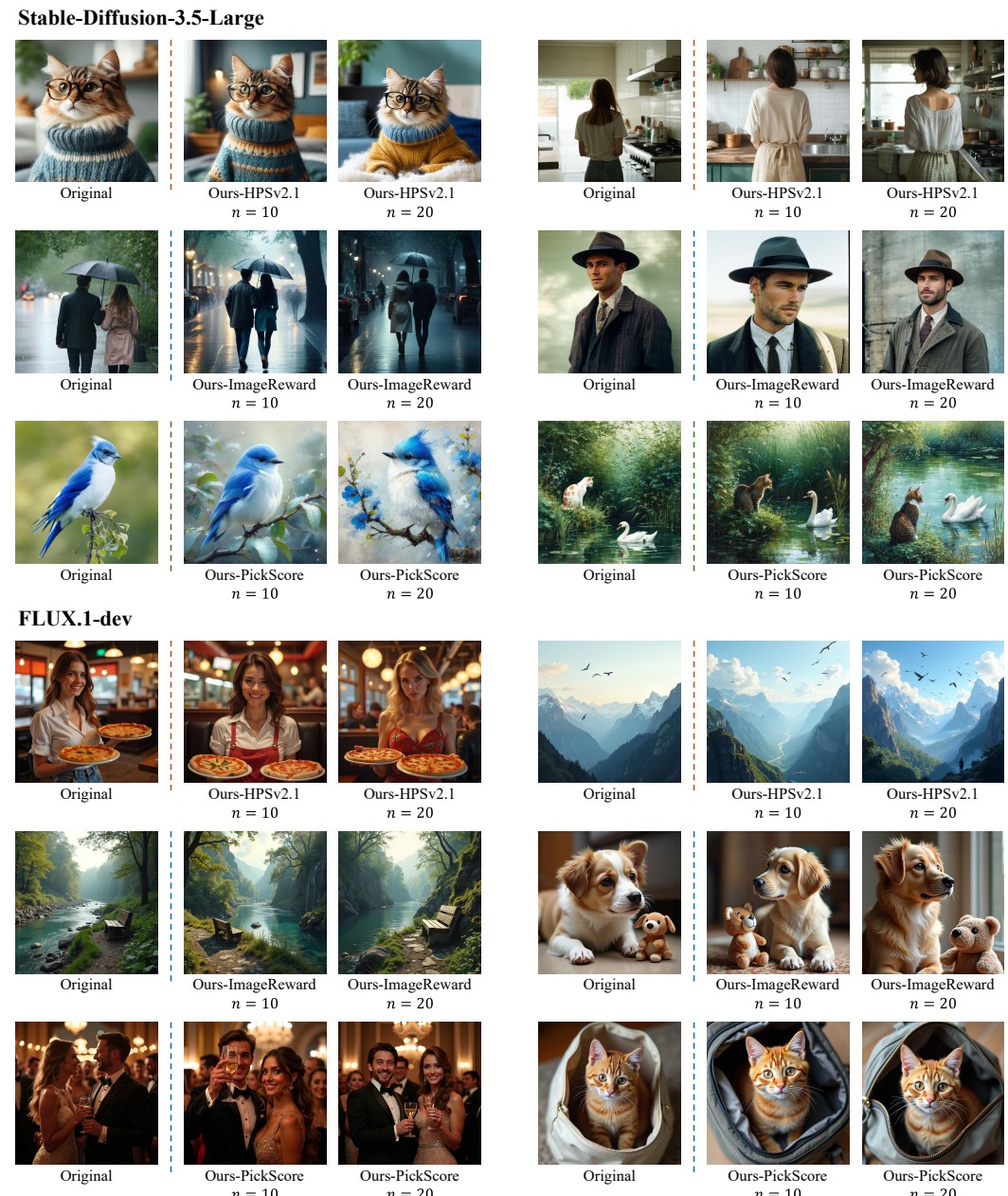

Figure 4: Visualization results of our proposed method. The images demonstrated here are of $1024 \times 1024$, and generated with prompts from COCO2014val dataset.

## A.6  ETHICS STATEMENT

This research follows the ethical guidelines established by the ICLR Code of Ethics. No experiments were conducted on human participants, and no private or sensitive data was collected or used. All datasets employed in our study are openly accessible and accompanied by appropriate licenses. While our work aims to advance the efficiency of diffusion model for inference-time scaling, we acknowledge that such techniques may be misapplied in harmful or malicious contexts. We therefore advocate for responsible use, in accordance with ethical norms and applicable legal frameworks.

### A.7 REPRODUCIBILITY STATEMENT

To improve reproducibility, we provide thorough documentation of our methodology. The main content and Appendix detail the model architectures, hyperparameters, and experimental setups. All datasets are publicly released. Moreover, we provide pseudocode, inference scripts, and implementation details to simplify replication efforts. Our source code is also provided in the supplementary material to ensure faithful reproduction of results.

### A.8 THE USE OF LARGE LANGUAGE MODELS

In this paper, large language models are only used to correct grammar and spelling errors.

