# OpenReview forum: "Sketch First, Scale Fast: On Efficient Inference-Time Scaling for Visual Generation"
_ICLR.cc/2026/Conference — ICLR 2026 Conference Withdrawn Submission_

### Official Review · Reviewer_YUNa · 2025-10-30

**Soundness:** 2
**Presentation:** 2
**Contribution:** 2
**Rating:** 4
**Confidence:** 5

**Summary:**

The paper proposes an efficient inference-time scaling method that allows scaling the number of candidates for choosing the candidate with maximum reward. It introduces a co-fusion module to identify the artifacts in the image early in the generation process and use a VLM verifier to get the feedback prompt to continue the generation towards the desired quality. Experiments are conducted on COCO val2014 dataset prompts using four different reward scores across SD3.5 and Flux models.

**Strengths:**

The proposed CoFusion module enables faster sampling by using a distilled Turbo variant to offload the computation of inference-time scaling achieving 2x speedup.

The visual reflection strategy enables to correct the model’s generation trajectory for better alignment with the reward function metric.

Experiments are conducted across four different metrics and two different models.

**Weaknesses:**

The proposed method assumes that the distilled version of the original models are available and follows the same trajectory as the base models. It is not applicable to models without  distilled variant. For a given seed, are the images generated by the original and distilled models identical? So, it is unclear if the feedback from the VLM is actually useful?

The benefits of the proposed method over the distilled model version are unclear. What’s the performance when the distilled versions are scaled with BoN or SDS? This baseline comparison is missing.

What does the feedback from VLM look like? It is better to show a few qualitative results for the examples visualized in Fig. 3 and 4.

The VLM still uses the noisy image obtained with Eq (12) to compare with the candidate image generated by the turbo variant and generate the feedback prompt. However, the VLM is not trained with such noisy images and so the feedback is bound to be error prone.

Does the method work with other metrics such as attribute binding or counting scores to improve the adherence to such prompts?

The method trades-off memory for computational cost by storing the intermediate noisy latents and so memory cost should also be discussed.

The paper writing can be improved. Figures are missing the prompt inputs, equations are not fully described, and the text has few typos. Figure 1 and 2 are not referenced in text. Proposition 1 is missing the statement. The theoretical analysis or proof can be moved to appendix since the method is not motivated from it, and it is added only to support the empirical results.

Flux model shows only marginal improvement with the Visual reflection module added. It is sensitive to the quality of the distilled model. Is it because of the errors in the distilled variant?

Qualitative results show only simple prompts where the original baseline already performs well. How does the method work for complex prompts?

Minor:
Typo in L61 (continue), L199 (delta_t_j), L315 (fast)

**Questions:**

See weaknesses above.

How does k and z impact the efficiency-accuracy tradeoff? An ablation study that can illustrate the gains at different values of k can explain it.

---

### Official Review · Reviewer_r9ZH · 2025-10-30

**Soundness:** 2
**Presentation:** 1
**Contribution:** 1
**Rating:** 2
**Confidence:** 4

**Summary:**

The paper introduces SF2, a framework for efficient inference-time scaling in diffusion-based image generation. The method decomposes the denoising process into two coordinated stages: candidate generation and visual reflection or refinement. In the first stage, the model generates multiple candidate sketches using a diffusion model initialized from a noise prior. Each candidate is evaluated by a reward function that quantifies prompt alignment or aesthetic quality. In the second stage, visual reflection is performed, where the diffusion model predicts a output conditioned on the current state. Then a vision-language model provides feedback on imperfections or inconsistencies, which is used to revise the prompt and guide the subsequent refinement process. The authors argue that this two-phase design enables SF2 to produce high-quality, semantically consistent images while substantially reducing inference cost. By exploiting cross-model interaction and language-guided feedback, SF2 achieves efficient and adaptive scaling of diffusion inference.

**Strengths:**

I find the motivation of the paper clear and well-justified. Efficient inference-time scaling is both interesting and important problem, and the authors provide a reasonable categorization of prior work into two directions: noise refinement and candidate pruning. This framing helps contextualize their contribution within existing literature. The experimental validation is also convincing, as the method is evaluated on two large-scale diffusion models, demonstrating its practical applicability and robustness in real-world generative settings.

**Weaknesses:**

Baseline comparison.  The choice of baselines is limited and does not adequately reflect the current state of research on inference-time scaling. The paper primarily compares SF2 with SDS and Best-of-N, both of which are relatively dated and represent only a narrow subset of existing methods. More recent and advanced scaling approaches [1–4] should have been included to ensure a fair and comprehensive evaluation of the proposed framework. Furthermore, the reported improvements over these baselines are relatively marginal, raising concerns about whether the additional computational overhead introduced by SF2 is justified given the limited performance gains.

Theoretical assumptions. The theoretical analysis relies on an assumption that the discrepancy between the base and turbo diffusion models is bounded by an L-Lipschitz constant. However, the paper does not provide empirical or theoretical evidence that this assumption holds in realistic scenarios. In practice, the two models may differ in dataset, training method, architecture, or source–target pairs, all of which could affect the Lipschitz constraint. Without a concrete justification or sensitivity analysis, the proof remains largely formal and may not meaningfully describe the empirical behavior of the system.

Presentation quality. The presentation of the paper could be significantly improved. Several notations are introduced without proper definition—for example, the symbols e, the superscript asterisk (*), and the delta with or without subscripts appear without clear explanation. This lack of clarity makes it difficult to follow the mathematical derivations, especially in the proof section, which becomes hard to parse and verify. More careful exposition, consistent notation, and a dedicated notation table would make the paper substantially more readable and rigorous.


[1] Nanye Ma, Shangyuan Tong, Haolin Jia, Hexiang Hu, Yu-Chuan Su, Mingda Zhang, Xuan Yang, Yandong Li, Tommi Jaakkola, Xuhui Jia, & Saining Xie (2025). Inference-Time Scaling for Diffusion Models beyond Scaling Denoising Steps. URL https://arxiv. org/abs/2501.09732.

[2] Kim, S., Kim, M., & Park, D. (2025). Test-time alignment of diffusion models without reward over-optimization. arXiv preprint arXiv:2501.05803.

[3] Luca Eyring, Shyamgopal Karthik, Karsten Roth, Alexey Dosovitskiy, & Zeynep Akata (2024). ReNO: Enhancing One-step Text-to-Image Models through Reward-based Noise Optimization. Neural Information Processing Systems (NeurIPS).

[4] Hwang, J., Kim, J., & Sung, M. (2025). Moment-and Power-Spectrum-Based Gaussianity Regularization for Text-to-Image Models. arXiv preprint arXiv:2509.07027.

**Questions:**

Please see weaknesses.

---

### Official Review · Reviewer_6yFT · 2025-10-31

**Soundness:** 1
**Presentation:** 1
**Contribution:** 2
**Rating:** 2
**Confidence:** 2

**Summary:**

This paper proposes an inference time scaling method (called SF2) which injects feedback from a vision language model (VLM) at an early stage during a text-to-image diffusion process. The authors justify the motivation as two-fold: (1) to improve alignment with human preferences and (2) to improve time-scaling methods by generating high quality pool of samples within limited budget. The authors claim that (1) their human alignment is superior thanks to a feedback from the VLM and (2) their method (called `Co-Fusion') can generate higher quality images while adhering to the compute budget (NFE). The method is verified on SOTA diffusion models: Stable-Diffusion-3.5-Large and FLUX.1-dev, on prompts from COCO2014val dataset. The performance estimators: HPSv2.1, ImageReward, PickScore and Aesthetic Score are averaged to show improvement on BoN (best-of-N) sampling and original diffusion methods.

**Strengths:**

This paper proposes a method to combine the advantages of human preference alignment within inference time scaling.

**Weaknesses:**

The paper is poorly written and hard to follow. Despite the experimental results, the theoretical analysis leaves many open questions which does not justify the claims.

Some suggestions of things to improve the paper that did not impact the score:
* noise vectors various markedly >> noise vectors vary markedly
* What is the metric $R$ in Eqn. (1) that is used to evaluate the samples and select a high-fidelity candidate? This should also be mentioned in Section 3.3
* visual of the outputs >> visual quality of the outputs

The authors argue that the Co-Fusion process generates samples of higher quality than BoN (and other stepwise trajectory search inference time scaling methods) by showing an error bound between the sample generated in step $N$ and the typical denoising process that takes $T$ steps, with $N<<T$. The step $N$ is chosen by denoising the image for the first $k$ steps with the original method followed by $S-z$ steps by the Turbo model. It is not clear how the final error bound presented in Proposition 1 is indeed small. The paper is difficult to follow with several undefined terms and expressions. I shall detail these in the comments below.

**Questions:**

* As the original denoising process is defined as $G(x(i), \theta)$, it is quite confusing to introduce the term $v_\theta$ in Eqn (3) without any reference to the original process. I assume $v_\theta$ is the partial derivate of this process at timestep $t_i$ and considering the input to the process at timestep $t_0$ is $x_{\theta, t_0}$.

* How can we conclude that $N<T$? Specifically, what is the magnitude of $k$ and $z$ to ensure this? A range of values could be provided for the models (and their respective turbo versions) considered.

 * In Proposition 1, what is $\mathbf{e}_{\theta, 0}$ in Eqn (4)? I assume it is $ x_{\theta, t_i} - x^*_{\theta, t_i}$. I assume $x^*$ is the image generated by the Co-Fusion process? I would recommend introducing clearly the notation for (1) the original process (2) the Turbo process and (3) the Co-Fusion process before beginning with the errors in Eqn (4).

* The distilled Turbo model is not introduced. What is $v_\psi$?

* The computations that follow Proposition 1 are correct. However, the error bound on $|| \epsilon_{\theta, i}||$ is not explained and appears suddenly in Eqn (4). What is $C$? Similarly $||\zeta||$ in Eqn (8) and the bound for it $\epsilon_{dis}$. As no details are provided, it is difficult to say whether these quantities are reasonably small. Since they strongly relate to the final bound in Eqn (11), it is hard to say whether the sample images chosen by the Co-Fusion process are indeed of better quality than the samples drawn from the original process at the time step $N$.

* The mathematical machinery in Proposition 1. is the same as in Appendix A1. So I do not understand why it is introduced as `supplementary material'.

---

### Official Review · Reviewer_5nLT · 2025-11-01

**Soundness:** 2
**Presentation:** 2
**Contribution:** 2
**Rating:** 4
**Confidence:** 4

**Summary:**

This paper addresses the inefficiency of existing inference-time scaling methods for diffusion models— which rely on redundant full sampling to generate candidate images—by proposing SF2, a novel efficient framework. SF2 consists of two core components: (1) The Co-Fusion pipeline, which accelerates candidate generation by combining early denoising steps from the original model (to preserve layout) with fast refinement via a distilled "Turbo" model (reducing total steps, $N=k+S−z<T$ where $T$ is the original model’s steps). This enables early candidate selection without sacrificing visual consistency. (2) The Visual Reflection mechanism, which leverages a vision-language model (Qwen2.5-VL) to identify flaws in selected candidates (e.g., distorted limbs) and injects feedback into the continued denoising process to correct artifacts.

**Strengths:**

1. Co-Fusion Balances Speed and Visual Consistency
   1. Early-step alignment preserves layout. and turbo model reduces steps without quality loss.
   2. Hyperparameter tuning optimizes trade-offs.
2. Visual Reflection Actively Corrects Artifacts
   1. VLM feedback targets specific flaws. This improves quality by addressing baseline limitations (selecting imperfect candidates).
   2. Feedback injection is seamless.
   3. Generalizes to video. Experiments on CogVideoX-2B shows Visual Reflection improves Visual Quality from 0.4859 (w/o) to 0.5264 (w/), confirming cross-modality applicability.
3. Efficient and Compatible Design
   1. Minimal runtime overhead. Table 2 shows SF2 adds 92–98s (Stable-Diffusion-3.5-Large) vs. BoN’s 198–204s, and Table 4 (Appendix A.2) reports Turbo models take 8/4 steps (vs. original 28), reducing per-candidate time.
   2. Compatible with existing models/samplers. Table 6 (Ours+DBCache) achieves HPSv2.1=34.01 (vs. Ours+Turbo’s 33.91), and Table 7 (video) validates extension to non-image models. No model modifications are needed—SF2 operates via denoising step substitution.
   3. Scales with candidate count. Table 1 shows SF2 (n=20) achieves higher Avg RI (7.80%) than n=10 (6.83) while using the same 280 NFEs as BoN (n=10), demonstrating efficient scaling.
4. Comprehensive Empirical Validation
   1. Cross-model consistency. Table 1 confirms robustness across diffusion architectures.
   2. Multi-metric evaluation. Table 1 (HPSv2.1, ImageReward, PickScore, Aesthetic Score) and Table 5 (Appendix A.3.1, MUSIQ, MAN-IQA, TOPIQ) show SF2 outperforms baselines across quality dimensions, avoiding metric bias.
   3. Ablations isolate component value. Table 3 shows Co-Fusion contributes 5.71% Avg RI, Visual Reflection adds 1.12% more, and their combination yields 6.83%—validating both components’ necessity.

**Weaknesses:**

1. Incomplete Analysis of Hyperparameters and Generalization
    1. k and z selection lacks systematic study. Figure 2 tests $k\in{0,4,5,7,16}$ but does not explain why k=7,z=3 (default) is optimal for all prompts. No ablations on how k/z impact complex prompts (e.g., "a busy street with 10 people") are provided.
2. Limited Evaluation of Content Preservation and Edge Cases
   1. No assessment of prompt alignment loss. Table 1 reports CLIP-like metrics (ImageReward) but no direct prompt alignment tests (e.g., BLIP-2 scores) for complex prompts. It is unclear if Visual Reflection preserves prompt intent while correcting artifacts.
   2. Edge cases (e.g., low-data prompts) are untested. The paper uses only common COCO prompts. No experiments on rare concepts, where Co-Fusion may fail to preserve layout.
   3. User study for aesthetics is missing. Aesthetic Score (LAION-Aesthetics) is used, but no user evaluations of "naturalness" (e.g., preference between SF2 and BoN) are conducted. Quantitative metrics may not capture subjective quality.
3. Reproducibility Gaps for Turbo Models and VLM Feedback
   1. VLM feedback generation is underspecified. The paper uses Qwen2.5-VL but does not provide the instruction prompt (p) used to generate feedback (e.g., "List flaws in this image: [image]"). Without this, users cannot replicate the feedback injection. Also, the robustness to VLM errors (e.g., false positives in flaw detection) is not evaluated.Erroneous feedback from VLM could degrade quality.
   2. Batch processing is unaddressed. All experiments use single-prompt generation, but no details on batch-wise Co-Fusion (e.g., batching candidate generation) are provided—limiting scalability for large datasets.
4. Incomplete Discussion of Computational Overhead
   1. The runtime of the VLM feedback step is omitted from total latency (Table 2; Sec. 4.2; p.7), despite Qwen2.5-VL-7B being non-negligible on A100 (Appendix A.2; p.15).
   2. Memory consumption of storing intermediate latents for n candidates is not reported, which could bottleneck large n (Algorithm 1; p.6).

**Questions:**

1. **How do k and z perform across diverse prompts (simple/complex, rare/common), and what guidance can be provided for tuning them?** The paper uses fixed
k=7,z=3 (Stable-Diffusion-3.5-Large) but does not test their impact on complex prompts or rare concepts (e.g., "a quokka holding a flower"). Could you add a supplementary table showing LPIPS, ImageReward, and runtime for some k and z on some diverse prompts? Additionally, could you provide a heuristic (e.g., "increase k for prompts with detailed layouts")?
1. **Does Visual Reflection preserve prompt intent while correcting artifacts, and how can this be quantified?** The paper reports ImageReward but no direct prompt alignment tests. Could you add experiments using BLIP-2 or CLIP scores to compare prompt alignment between SF2, BoN, and the original model? Additionally, could you include qualitative examples where Visual Reflection might alter prompt-critical details (e.g., changing a "blue apron" to "red")?
2. **Can you provide ablations study on VLMs?** Could you share the instruction prompt (p) used to generate VLM feedback (e.g., "Identify 1–2 flaws in this image, e.g., distorted limbs, low fidelity") and an example feedback string? Also, one VLM is not sufficient to validate the robustness of Visual Reflection. Could you test another VLM (e.g., DeepSeek-VL, LLaVA) to see if feedback quality impacts results?
4. **How does SF2 perform on diversity data (prompts)?** Experiments use only COCO prompts (To be specific, they are image captions only, not those useful and well-tested prompts that the current T2I community is using and those prompts are more suitable for the 2 models in your paper), which is far away from sufficient to validate the generalization of SF2 in the text-to-image generation task. Could you add experiments on more diverse datasets like DrawBench (with abstract/creative prompts), and PartiPrompts (with compositional prompts)? This would clarify if SF2 generalizes beyond common prompts.

Overall, although using VLM to guide and select candidates is widely explored by previous works, the Co-Fusion proposed in this paper is novel. However, my main concern is the testing prompts of your experiments, which is not quite standard in the current T2I research field and the main drawback of this paper. I would consider raising the score only if the authors provide experiments on more diverse and challenging prompts.

---

### Note · Authors · 2025-12-02

I have read and agree with the venue's withdrawal policy on behalf of myself and my co-authors.